# PepTriX: A Framework for Explainable Peptide Analysis through Protein Language Models

## Abstract

Peptide classification tasks, such as predicting toxicity and HIV inhibition, are fundamental to bioinformatics and drug discovery. Traditional approaches rely heavily on handcrafted encodings of one-dimensional (1D) peptide sequences, which can limit generalizability across tasks and datasets. Recently, protein language models (PLMs), such as ESM-2 and ESMFold, have demonstrated strong predictive performance. However, they face two critical challenges. First, fine-tuning is computationally costly. Second, their complex latent representations hinder interpretability for domain experts. Additionally, many frameworks have been developed for specific types of peptide classification, lacking generalization. These limitations restrict the ability to connect model predictions to biologically relevant motifs and structural properties. To address these limitations, we present PepTriX, a novel framework that integrates one dimensional (1D) sequence embeddings and three-dimensional (3D) structural features via a graph attention network enhanced with contrastive training and cross-modal co-attention. PepTriX automatically adapts to diverse datasets, producing task-specific peptide vectors while retaining biological plausibility. After evaluation by domain experts, we found that PepTriX performs remarkably well across multiple peptide classification tasks and provides interpretable insights into the structural and biophysical motifs that drive predictions. Thus, PepTriX offers both predictive robustness and interpretable validation, bridging the gap between performance-driven peptide-level models (PLMs) and domain-level understanding in peptide research.

## 1 Introduction

Peptides De La Rica & Matsui (2010), short chains of 2–50 amino acids, are central to diverse biological processes including hormone regulation, cell signaling, antimicrobial defense, and therapeutic activity Wang et al. (2022a). Their biological properties arise from the primary amino acid sequence and the resulting three-dimensional (3D) structure. Accurate peptide classification is critical for predicting properties such as toxicity, receptor binding, antimicrobial activity, and therapeutic potential. However, experimental characterization remains costly and time-consuming, making computational prediction indispensable for large-scale peptide discovery Zhou et al. (2016); Goles et al. (2024). Early computational methods relied on manually engineered features such as amino acid composition, physicochemical descriptors, or substitution matrices, combined with classical machine learning algorithms (e.g., SVMs (Sanders et al., 2011a), Random Forests (Fusaro et al., 2009)). While these approaches provided useful baselines, they were limited in scalability and struggled to capture the complex, nonlinear sequence–function relationships inherent to peptides. Recent advances in deep learning and protein language models (PLMs) have transformed peptide and protein prediction. By leveraging self-supervised training on millions of protein sequences, PLMs encode evolutionary and biochemical information directly from raw sequence data. Transformer-based models such as ProtBert (Brandes et al., 2022) and ESM-2 (Lin et al., 2023a) have advanced functional classification, while AlphaFold2 (Jumper et al., 2021a) and ESMFold (Lin et al., 2023a) have revolutionized 3D structure prediction, achieving near-experimental accuracy at scale. These developments have accelerated peptide classification by linking sequence with structural context. Despite these advances, interpretability and very high demand of computational resources remains a major

challenge. The practical application is significantly constrained by two factors: first, the substantial computational costs associated with their operation; and second, their intrinsic nature as a "black box," which obscures the reasoning behind their predictions. This absence of transparency is a direct consequence of their architectural complexity Hunklinger & Ferruz (2025). PLMs are characterized by the implementation of a deep stack of layers, with each layer being equipped with a multitude of multi-head attention mechanisms Vig et al. (2021). This configuration leads to the generation of hundreds of individual attention maps. In contrast to a single, focused lens, this design functions as a multitude of overlapping, abstract perspectives operating in parallel. The specific biological signal for a given task, such as a toxic motif cannot be readily differentiated in a single layer. Instead, it is fragmented and dispersed throughout this intricate network of patterns Ghotra et al. (2021). Trying to sort through these hundreds of attention scores to uncover a clear biological cause for a prediction becomes a challenging post-hoc analysis effort, frequently yielding ambiguous results.

Several recent works have addressed interpretability in peptide and protein models. PepNet Han et al. (2024) is framework employs a combination of residual convolutional and transformer blocks, along with pre-trained protein language model embeddings, to make predictions. The interpretability of the model is facilitated by saliency maps, which underscore significant regions within the peptide sequence. However, a notable constraint pertains to the potential misinterpretation of these saliency maps to encapsulate the comprehensive decision-making process of the model. Also, the CNNs fail to capture long-term interdependency, leading to inaccurate predictions for peptides with very long amino acid chains (>20 amino acids) Lei et al. (2021). The iCAN Weckbecker et al. (2024) framework utilizes a carbon-neighborhood array encoding methodology to represent molecular structures, which can subsequently be employed for prediction tasks. iCAN offers interpretability by generating feature relevance heatmaps that show the importance of different molecular features. A primary limitation of this approach is its focus on local chemical neighborhoods, which causes it to miss important global patterns in the peptide sequence, and not explain which residues drive predictions Liu et al. (2025). PHAT Jiang et al. (2023) is a framework that utilizes a deep hypergraph attention network to model the secondary structure and discriminative substructures of short peptides. The interpretable output of this model consists of substructure-level feature maps that visualize the segments driving the prediction. The primary constraints imposed by PHAT pertain to the intricacy inherent in constructing hypergraphs and the model's tendency to overfit, a phenomenon that is particularly salient when dealing with modest datasets. The PLPTP (Peptide-Ligand-Target Prediction) Gao et al. (2025) framework integrates evolutionary and structural features using ESM-2, a Bidirectional LSTM, and motif mining for peptide property and toxicity prediction. The program provides interpretability at the motif level, attributing predictions to specific sequence motifs. A significant constraint pertains to its reliance on predefined motifs, which lacks the capacity to identify novel or previously unobserved patterns in peptide sequences. BERT-NeuroPred Singh et al. (2024) is framework adapts the BERT model for peptide prediction by combining a BERT Devlin et al. (2019) encoder with genetic feature selection. The system's interpretability is facilitated by attention mechanisms and statistical tests, which are employed to identify significant residues. The limitations of this framework include the risk that attention weights may not accurately reflect the true importance of features and the high computational cost associated with using large-scale Transformer models Wen et al. (2022). Umami-BERT Zhang et al. (2023) is a framework that utilizes the capabilities of BERT for umami peptide prediction by employing sequence embeddings. The model offers residue-level attention importance as its interpretable output. As with other BERT-based models, potential limitations include the possibility of misleading attention-based explanations and the significant computational resources required to train and run the model. Multi-Target Quantitative Structure-Activity Relationship (MLT-QSAR) Liu et al. (2011) is a framework of regression and classification models that relate the physicochemical properties of a substance to its biological activity. The interpretability of QSAR models derives from their capacity to ascertain the contribution of each descriptor to the model's prediction. A significant constraint of QSAR de Tilleghem & Govaerts (2007); Soares et al. (2022) is its reliance on manually crafted features, which does not fully capture the intricacies of non-linear relationships or novel sequence patterns, and the feature engineering is often time-consuming and tedious. QSAR has shown to provide remarkable performance for HIV datasets but it is not generalizable to other datasets. InterPLM applied sparse autoencoders to extract human-interpretable features from ESM-2 (Simon & Zou, 2024), demonstrating that PLMs capture known biological concepts, though not directly for peptide function. Wenzel et al. fine-tuned ProtBert Brandes et al. (2022) and ESM-2 for GO term and enzyme annotation and extended integrated gradients to identify sequence regions influencing predictions (Wenzel

et al., 2024). However, these methods either lack broad benchmarking across peptide classification tasks or do not fully integrate sequence and structural determinants. Most of the frameworks discussed here such as PLTP (for toxicity), QSAR (for HIV), BERT-NeuroPred (for neuropeptide), Umami-BERT (for umami peptides) are not generalizable to all datasets and are limited to the classification of specific types of peptides. All of the frameworks require extensive parameter tuning, which leads to results that vary with the choice of hyperparameters. Models based on engineered features (physicochemical, motif-based) can fail to capture hidden or novel patterns, may overlook crucial information, and often require expert domain knowledge, reducing adaptability to atypical peptides or unknown classes.

In this work, we address these challenges by proposing a framework that integrates one-dimensional (1D) sequence and three-dimensional (3D) structural representations for peptide classification. Our approach leverages ESM-2 embeddings for sequence information and ESMFold-derived structures refined through a graph attention convolution network (GAT) to detect biochemically relevant motifs. Using contrastive learning, we align sequence and structure modalities to produce predictive and interpretable peptide embeddings through a hybrid-loss function. Our contributions are threefold: (i) We benchmark our framework on diverse peptide classification tasks, achieving comparable or improved performance compared to existing methods; (ii) we demonstrate that cross-modal embeddings generalize across different biological domains through a hybrid-loss function; and (iii) we provide interpretability by linking sequence features to structural determinants of peptide function. Together, these advances address the dual challenge of accuracy and explainability in peptide classification.

## 2 METHODOLOGY

In this section, we will discuss our methodology and framework. We justify the methodology by demonstrating how it overlaps with the fields of artificial intelligence and bioinformatics (Figure 1).

### 2.1 EVOLUTIONARY SCALE MODELING (ESM)

ESM-2 Lin et al. (2023a) is a Transformer-based protein language model developed by Meta AI, trained on 138 million UniRef90 Consortium (2007) protein sequences to understand biology's semantics, including evolutionary patterns, contextual relationships, and amino acid dependencies. As the model scales to 15 billion parameters, it encodes biological context and structural information up to atomic resolution, revealing that protein sequences contain subtle signals embedded by evolutionary pressures Cagiada et al. (2023). ESM-2 leverages self-attention Shaw et al. (2018) to capture such long-range and complex dependencies, producing rich embeddings for each amino acid. These embeddings represent deep contextual and evolutionary meaning, enabling downstream tasks such as structure prediction and functional annotation. ESMFold is a generative deep learning model developed by Meta AI Lin et al. (2023a), designed to predict the three-dimensional atomic structure of a protein directly from its 1D amino acid sequence. The ESM-2 protein language model is used to create sequence embeddings that are translated into structural coordinates. The sequence is encoded by ESM-2, producing contextual representations for each residue. These are processed through folding blocks, refinement of residue-level and pairwise features, and passed to an equivariant Transformer-based structure module for accurate 3D atomic coordinates and calibrated confidence estimates Lin et al. (2023a). The peptide's function is fundamentally determined by its 3D conformation; access to structural information is essential. Traditional experimental methods, such as X-ray crystallography, while accurate, are costly, time-intensive, and not scalable to the vast diversity of proteins and peptides Iglesias et al. (2025). ESMFold Lin et al. (2023b) addresses this bottleneck by providing a fast, accurate, and high-throughput approach to 3D structure prediction. By efficiently transforming the 1D amino acid sequence into its corresponding 3D fold, ESMFold enables large-scale multimodal analyses and serves as an essential preprocessing step as input for our Graph Attention Network (GAT). We selected ESMFold for its fast, alignment-free inference from single sequences, which streamlines our pipeline and enables high-throughput peptide modeling; this makes it a strong practical choice alongside multiple-sequence-alignment–based predictors such as AlphaFold2 Nature Methods (2023); Jumper et al. (2021a). This makes it better suited for high-throughput applications like ours and does not tie its usage to access large amounts of computational resources.

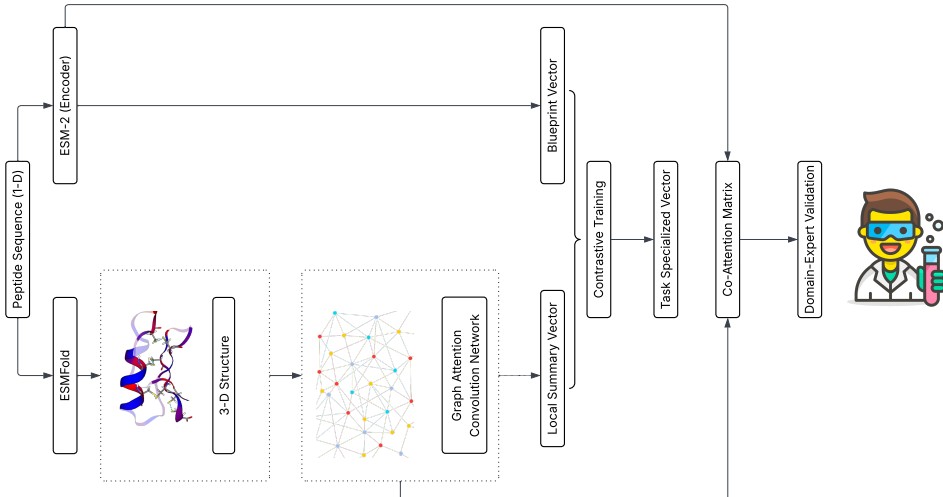

Figure 1: The framework predicts peptide function by integrating a one-dimensional sequence, or blueprint vector, and a three-dimensional structure, or local summary vector. It uses a dual-encoder architecture to learn a shared representation of these two modalities. The one-dimensional sequence is processed using a protein language model (ESM-2) to capture contextual and evolutionary features. Meanwhile, the three-dimensional structure is generated using a model like ESMFold and analyzed by a graph attention convolution network (GATConv). Contrastive training ensures alignment between the vectors. A co-attention module generates a co-attention matrix for domain expert validation. GATConv specializes in identifying unique structural motifs relevant to a specific dataset.

## 2.2 GRAPH ATTENTION NETWORK (GAT)

A graph convolutional neural network (GCN) Kipf & Welling (2017) is a type of neural network designed to process structured graph data. GCNs can learn from this type of data, making them useful for applications involving molecular structures, social networks, and recommendation systems, among others. GCNs operate through a process involving message parsing. In the context of peptide sequences, a GCN can be understood as a model that analyzes and updates its feature vector by examining each amino acid (node) and its neighbors. However, in its basic form, the GCN averages information from its neighbors McDonnell et al. (2022); Garzon et al. (2024), which is one of its drawbacks. For example, if we consider an aspartic acid (ASP) residue in the binding pocket, we may have neighbors such as a lysine (LYS), which forms a charge-based salt bridge, an important component; a phenylalanine (PHE) residue, which contains bulky rings that help form the pocket's shape; and a glycine (GLY) residue, which is small and neutral Ma et al. (2003). A vanilla GCN would consider all of the aforementioned residues equally important Xia et al. (2021). However, in terms of peptide relevance, the interaction with LYS is much more important than the interaction with GLY Gao & Skolnick (2012); Stank et al. (2016). Therefore, we use a graph convolution attention network (GAT) Veličković et al. (2018); Chen et al. (2024), which learns to focus on the most biochemically significant interactions rather than treating all neighbors equally Wang et al. (2022b).

A single GATConv layer updates a central node's feature vector, $\vec{h}_i$, by performing the following steps:

1. **Feature Transformation:** All node features in the neighborhood $\mathcal{N}_i$ of node $i$ are transformed by a shared weight matrix $\mathbf{W}$ (Equation: 1).

$$\vec{h}'_j = \mathbf{W}\vec{h}_j \quad \forall j \in \mathcal{N}_i \cup \{i\} \tag{1}$$

$$e_{ij} = \text{LeakyReLU}\left(\vec{a}^T[\mathbf{W}\vec{h}_i \| \mathbf{W}\vec{h}_j]\right) \tag{2}$$

   Prior to the initiation of message passing, the feature vector of each node in the graph is processed through a standard linear layer, denoted by a weight matrix $W$. This process involves the projection of the initial, rudimentary features into a higher-level feature space,

thereby enabling the model to acquire more complex representations. The transformed feature for our aspartic acid is designated $h_{ASP}$, and its neighbors, $h_{LYS}$, $h_{PHE}$, and $h_{GLY}$, are likewise labeled.

2. **Attention Coefficient Calculation:** An non-normalized attention score Vaswani et al. (2017), $e_{ij}$ (Equation: 2), is computed for each neighbor $j$, signifying its importance to node $i$. This is achieved through the implementation of a single-layer feedforward network, parameterized by a weight vector, represented by the symbol $\vec{a}$. This network is applied to the concatenated transformed features. This fundamental principle constitutes the core of the GAT. For each neighbor of aspartic acid, the model calculates an "attention coefficient" ($e$) that represents the importance of that neighbor to aspartic acid. For the neighbor Lysine, the following procedure is employed: the transformed vectors $h_{ASP}$ and $h_{LYS}$ are concatenated, and they are then passed through a small, single-layer feed-forward neural network which is parameterized by a weight vector, $\vec{a}$. This network produces a single raw score, $e_{ASP, LYS}$. This process is repeated for all neighbors, resulting in the production of $e_{ASP, PHE}$ and $e_{ASP, GLY}$. These $e$ values represent the raw importance scores. A higher score indicates that the model has learned that this specific interaction is more significant. This consideration allows us to capture higher-order information at the amino acid level.

3. **Normalization:** The normalization of the attention scores across all the neighbors (amino acids) is achieved through the implementation of the softmax function which facilitates the creation of the comparable attention weights, denoted by $\alpha_{ij}$ (Equation: 3).

$$\alpha_{ij} = \text{softmax}_j(e_{ij}) = \frac{\exp(e_{ij})}{\sum_{k \in \mathcal{N}_i} \exp(e_{ik})} \qquad \vec{h}'_i = \sigma\left(\sum_{j \in \mathcal{N}_i} \alpha_{ij} \mathbf{W} \vec{h}_j\right)$$
$$(3) \qquad\qquad\qquad\qquad (4)$$

4. **Weighted Aggregation:** The final updated feature vector for node $i$, denoted by $/vech'_i$, is a weighted sum of its neighbors' features, utilizing the calculated attention weights. Subsequently, an activation function, $\sigma$ (Equation: 4)(GeLU Hendrycks & Gimpel (2016) or ReLU Nair & Hinton (2010)), is applied.

The final step in this process is to create the new, updated feature vector for our aspartic acid. Rather than employing a straightforward arithmetic mean, a weighted average is utilized, with the attention weights determined in the preceding step. $\vec{h}'_{ASP} = \alpha_{ASP,LYS} * \vec{h}_{LYS} + \alpha_{ASP,PHE} * \vec{h}_{PHE} + \alpha_{ASP,GLY} * \vec{h}_{GLY}$ The resulting vector, denoted as $\vec{h}'_{ASP}$, serves as a comprehensive representation of the aspartic acid's structural environment, considerably influenced by its predominant biochemical partners. The GAT uses multi-head attention to create a comprehensive, biochemically informed representation of the peptide's 3D structure. Iterative steps apply different learned weights, and the final outputs are aggregated to create a rich feature vector. Repeated for multiple layers, the GAT constructs an intricate understanding of the peptide's complete 3D conformation.

## 2.3 CONTRASTIVE TRAINING

The objective of contrastive learning Chopra et al. (2005) is to instruct the sequence encoder (ESM-2) and the structure encoder (GAT) in a shared, consistent language. This process is facilitated by the utilization of summary vectors derived from the output of each encoder, namely the sequence vector, denoted by $\vec{z}_{seq}$, and the structure vector, denoted by $\vec{z}_{struct}$. Contrastive training involves the utilization of two expert models operating in parallel: an ESM-2 encoder for one-dimensional (1D) sequences and a GAT encoder for three-dimensional (3D) structures. The training procedure uses "matchmaking" as the underlying model, recognizing sequences corresponding to specific structures. Positive pairs are similar, while dissimilar pairs are negative. The contrastive learning process is based on the attraction of positive pairs and the repulsion of negative pairs. The model generates a summary vector for each peptide sequence and structure in the batch, with a contrastive loss function evaluating these vectors and providing feedback. The function rewards the model for making sequence and structure vectors similar, pulling them closer together in a high-dimensional embedding and vice-versa. In the field of biochemistry, the one-dimensional (1D) amino acid sequence can be regarded as the blueprint vector, as it contains all the instructions for the formation of a peptide. The three-dimensional (3D) folded structure, in contrast, can be considered the completed structure or a summary vector. Contrastive learning is a method designed to force the model to learn the fundamental biophysical rules connecting the blueprint to the structure. The fundamental prerequisite

for the success of this model is the recognition that the biochemical properties of a sequence must be reflected in its final structure. Contrastive learning employs a loss function, typically InfoNCE (noise-contrastive estimation) Gutmann & Hyvärinen (2012), to achieve this objective. In the context of a given set of N peptides, calculating loss from the sequence's perspective consists of two primary components: the numerator (the positive pair score) and the denominator (the total score). Loss for a single pair is equivalent to the negative logarithm of the ratio of the two parts, analogous to a softmax function. From the structure's perspective, a symmetrical loss is calculated by comparing the structure vector of peptide i to all sequence vectors. The final contrastive loss for the batch is the average of these two symmetrical losses, ensuring bidirectional alignment. The training objective is based on Noise-Contrastive Estimation (InfoNCE). For a batch of $N$ peptides, we have $N$ positive pairs $(\vec{z}_{seq,i}, \vec{z}_{struct,i})$ and $N(N-1)$ negative pairs $(\vec{z}_{seq,i}, \vec{z}_{struct,j})$ where $i \neq j$. The contrastive loss for a single positive pair $i$ from the sequence perspective is given by Equation: 5.

$$\mathcal{L}_{i,seq} = -\log \frac{\exp(\text{sim}(\vec{z}_{seq,i}, \vec{z}_{struct,i})/\tau)}{\sum_{j=1}^{N} \exp(\text{sim}(\vec{z}_{seq,i}, \vec{z}_{struct,j})/\tau)} \quad (5) \qquad \mathcal{L}_{\text{contrastive}} = \frac{1}{2N} \sum_{i=1}^{N} (\mathcal{L}_{i,seq} + \mathcal{L}_{i,struct}) \quad (6)$$

where $\text{sim}(\cdot, \cdot)$ is the cosine similarity and $\tau$ is a temperature hyperparameter. A symmetrical loss, $\mathcal{L}_{i,struct}$, is calculated from the structure perspective. The total contrastive loss for the batch is the average over all pairs (Equation: 6). This loss necessitates that the encoders generate representations that are not only beneficial for the subsequent task but are also fundamentally aligned across the two modalities. To illustrate, a sequence characterized by an alternating pattern of cationic (positively charged) and hydrophobic (water-repelling) residues, such as K-L-A-K-K-L-A..., exhibits a high propensity to form an amphipathic alpha-helix Drin & Antonny (2010); Yin et al. (2012). This specific 3D structure is a critical feature for many antimicrobial peptides, as it allows them to insert into and disrupt bacterial membranes Tossi et al. (2000). This model posits that the sequence vector representing the K-L-A-K... pattern must be in close proximity to the structure vector representing an amphipathic helix. Concurrently, the model is instructed that this specific sequence vector should be distant from the structure vector of a globular, all-beta-sheet protein. Through this alignment, the model does more than memorize patterns; it develops a fundamental understanding of folding principles. Clearly, the information in the 1D blueprint is not arbitrary. Rather, it directly causes the final 3D functional form Rost et al. (1998); Jumper et al. (2021b). This approach enhances the internal reasoning's robustness, generalizability, and alignment with real-world biochemistry. The training of the entire network is achieved by minimizing a hybrid loss function, which is a weighted sum of the task-specific classification loss and the multi-modal contrastive loss given by $\mathcal{L}_{total} = \mathcal{L}_{classification} + \lambda \mathcal{L}_{contrastive}$ Wang et al. (2021); Gunel et al. (2020). Here, $\mathcal{L}_{\text{classification}}$ is the Binary Cross-Entropy Mao et al. (2023) loss that measures predictive performance, and $\lambda$ is a scalar weight that balances the two objectives. This approach is designed to ensure that the model learns to function as both an accurate predictor and a coherent multimodal reasoner. The hybrid loss function plays a central role in the training process. The initial component is a conventional classification loss that quantitatively evaluate predictive performance, enabling the model to identify the biological function of peptides, for instance, "antimicrobial." The second component, contrastive loss, enforces biochemical determinism by focusing on the principle that a peptide's sequence determines its structure. This demonstrates that the model can recognize that a sequence with a repeating cationic-hydrophobic pattern (the 1D blueprint) should have a vector representation that is mathematically similar to that of an amphipathic $\alpha$-helix (the 3D structural summary); a critical structure for membrane disruption Jumper et al. (2021b). The integration of these two error signals into a weighted sum compels the model to meet both objectives. It is not possible for the model to attain a low total error by only being accurate without comprehending the underlying biophysics. Similarly, it cannot prioritize internal consistency without making accurate predictions. This dual pressure is instrumental in ensuring that the model learns to function as a potent predictor, whose reasoning is firmly rooted in the multimodal scientific principles underlying the creation of a peptide's functional shape.

## 2.4 INTERPRETABILITY ANALYSIS THROUGH CROSS-MODAL ATTENTION

In this framework, the ESM-2 and GAT encoders can be conceptualized as two highly specialized experts conducting an investigation into a peptide. ESM-2 is the "linguist" who comprehends the 1D sequence, and GATConv is the "architect" who understands the 3D structure. While each report is valuable in its own right, they are best regarded as discrete entities. The Cross-Modal

Attention (or Co-Attention) module is the mechanism where these two experts engage in a deep, structured dialogue to create a single, unified theory. The Query-Key-Value (QKV) model is a powerful tool for identifying structural features in complex data. It involves two experts forming a query based on available information. For example, the ESM-2 analyzes a cysteine residue, asking which components are most relevant to its potential role. The other expert provides a searchable structural summary, with the GAT architect producing a key vector for each residue in the 3D structure. The value vectors provide comprehensive analysis, including all structural positions, containing extensive contextual information learned through the framework's implementation. This model has proven effective in various domains. The attention mechanism in GAT architecture involves comparing queries from experts, identifying relevant keys, and retrieving a weighted sum of corresponding values. This bidirectional dialogue allows the model to construct a comprehensive understanding of peptide function. The formation of a 3D functional epitope, a precise spatial arrangement of amino acid side chains, is the predominant factor in this process. A 3D functional epitope, such as the zinc finger motif, is a paradigmatic example of this, where distant cysteine and histidine residues fold to form a pocket that coordinates a zinc ion Krishna et al. (2003). The co-attention mechanism is a deep, layer-by-layer fusion technique that enables a comprehensive understanding of peptide function. It identifies non-local relationships between sequence and structure residues, with visual attention weights illustrating the strength of these connections. This allows for direct observation of the model's representation of the predicted functional epitope. Mathematically, the co-attention module implements the "scaled dot-product attention" mechanism. Given the sequence embeddings $\mathbf{S} \in \mathbb{R}^{n \times d}$ and structure embeddings $\mathbf{G} \in \mathbb{R}^{n \times d}$ for a peptide with $n$ residues, the dialogue is bidirectional. To make the sequence "structure-aware," the model learns three weight matrices $(\mathbf{W_Q}, \mathbf{W_K}, \mathbf{W_V})$ to project the inputs into the Query, Key, and Value spaces: $Q_S = SW_Q$, $K_G = GW_K$, $V_G = GW_V$

$$\text{Attention}(\mathbf{Q_S}, \mathbf{K_G}, \mathbf{V_G}) = \text{softmax}\left(\frac{\mathbf{Q_S K_G}^T}{\sqrt{d_k}}\right) \mathbf{V_G} \quad (7) \qquad \text{Attention}(\mathbf{Q_G}, \mathbf{K_S}, \mathbf{V_S}) = \text{softmax}\left(\frac{\mathbf{Q_G K_S}^T}{\sqrt{d_k}}\right) \mathbf{V_S} \quad (8)$$

The term $\mathbf{Q_S K_G}^T$ calculates the dot product between sequence Query and structure Key, producing raw similarity scores. The scaling is done using the square root of the diagonal element, $\sqrt{d_k}$, to ensure gradient stability. The softmax function transforms these scores into a probability distribution, represented by the attention weight matrix. The matrix is then multiplied by the structure's value matrix to derive a sequence embedding matrix i.e., $\mathbf{V_G}$. This matrix updates each residue's representation with a weighted sum of structural information. An exact, symmetrical process is performed to make the structure "sequence-aware" (Equation: 8). This block, often augmented with multi-head attention, is the primary driving force behind the integration of the two modalities.

## 3 RESULTS AND DISCUSSION

**Experimentation Setup**: The framework was trained and tested on a computing cluster equipped with 128 GB of RAM, two AMD Epyc 64-core processors, and two Nvidia A100 GPUs with 40 GB of vRAM. The sequences were divided into 32 batches for training, validation, and testing. A 1D sequence was used to create a blueprint vector and 3D structures for ESM-2 and ESMFold. The framework had 162 million trainable parameters and was trained for 10 epochs, stopping if the validation loss did not improve for consecutive 3 epochs. The objective was to train a co-attention interpretability mechanism to provide insights and generate a visualization for biochemical experts to validate the findings. Contrastive learning was implemented to ensure the robustness of the co-representation learned by the 1D and 3D sequences despite varying hyperparameters, as demonstrated by the Robustness Verification Framework for Contrastive Learning (RVCL) Wang & Liu (2023).

**Domain-Centric Interpretability Analysis**: We trained and evaluated our model on nine different two class datasets containing different types of peptides with a broad range of biological functions and structures (Table 1). The datasets involve the binary classification of various peptide types: anti-inflammatory (aip_antiinflam Gupta et al. (2017)), pro-inflammatory (pip_pipel Manavalan et al. (2018a)), anti-microbial (amp_iamp2l Xiao et al. (2013) and amp_modlamp Müller et al. (2017)), amyloidogenic hexapeptides (amy_hex Prabakaran et al. (2021)), HIV V3-loop tropism (X4 vs. R5) (hiv_v3 Dybowski et al. (2010)), insect neuropeptides (nep_neuropipred Agrawal et al. (2019b)), soluble E. coli peptides (sol_ecoli Prabakaran et al. (2021)), and toxic peptides (toxinpred_trembl Gupta et al. (2013)). For results on additional datasets, refer Table Appendix A.1. Training data points vary

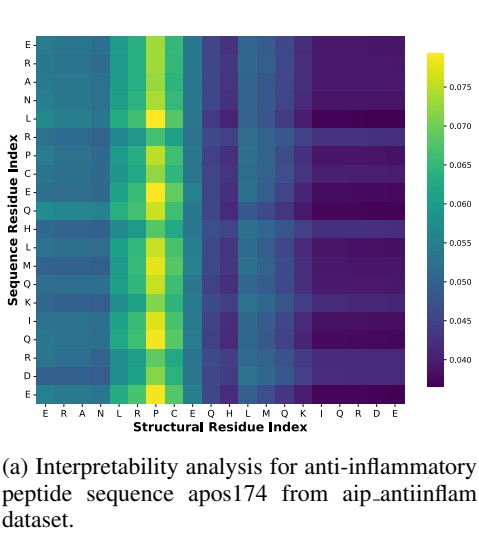

(a) Interpretability analysis for anti-inflammatory peptide sequence apos174 from aip_antiinflam dataset.

(b) Interpretability analysis for anti-microbial peptide sequence 723 from amp_modlamp dataset.





(c) Interpretability analysis for the R5 peptide sequence 266 from the hiv_v3 dataset.

(d) Interpretability analysis for the neuropeptide sequence 863 from the nep_neuropipred dataset.

Figure 2: Interpretability analysis for four different sequences from the datasets: aip_antiinflam, amp_modlamp, hiv_v3 and nep_neuropipred dataset. The heatmaps show the attention matrix learned by the classifier, with rows corresponding to sequence residues and columns to structural residue indices. Brighter colors show higher attention and darker colors lower attention.

between 700 to 10,041 sequence with a significant class imbalance, from highly balanced as in the amp_modlamp dataset to highly imbalanced as in the hiv_v3 dataset. The shortest peptide among the listed datasets comprises four amino acids (AA) and originates from the toxinpred_trembl dataset, whereas the longest peptide comprises 35 AA and also originates from the toxinpred_trembl dataset. The overall performance for the differentiation between classes is high with four datasets reaching F1-scores of 0.9 and higher. In this section, we discuss the interpretability results for a subset of the data we experimented our framework with due to text constraints. Figure fig:all depicts the attention of four peptides. The heatmaps show distinct patterns that correspond to the importance of individual amino acids within the structure, and these patterns can be interpreted. Figure 2a shows that the model pays the most attention to proline (P). This sequence is an anti-inflammatory peptide, which means that it regulates the immune system. Proline-rich motifs (PRMs) are often found in peptides and proteins involved in the immune response, which is why the peptide was positively identified. PRMs typically adopt a polyproline II helix conformation, which promotes intermolecular interactions such as signal transduction. This is why the model considers P to be the most important amino acid for structure and classification Srinivasan & Dunker (2012). Figure 2b shows that the model

Table 1: The performance metrics shown here are based on various datasets and include sample sizes, class balance, precision, recall, and F1-scores for classes 0 and 1. The F1-scores in bold either match or exceed the state of the art, which refers to vector encodings that work with diverse datasets. See Table A.1 for performance on those datasets.

| Dataset | Train | Test | $Bal._0$ | $Bal._1$ | $Prec._0$ | $Prec._1$ | $Rec._0$ | $Rec._1$ | F1-Score |
|---|---|---|---|---|---|---|---|---|---|
| aip_antiinflam | 1486 | 425 | 0.61 | 0.39 | 0.77 | 0.61 | 0.73 | 0.67 | **0.70** |
| amp_iamp2l | 2298 | 657 | 0.69 | 0.31 | 0.92 | 0.95 | 0.98 | 0.81 | **0.93** |
| amp_modlamp | 1805 | 516 | 0.50 | 0.50 | 0.95 | 0.96 | 0.97 | 0.95 | **0.96** |
| amy_hex | 994 | 285 | 0.64 | 0.36 | 0.86 | 0.76 | 0.87 | 0.75 | **0.83** |
| hiv_v3 | 945 | 271 | 0.87 | 0.13 | 0.99 | 0.84 | 0.97 | 0.91 | 0.97 |
| nep_neuropipred | 1225 | 350 | 0.51 | 0.49 | 0.89 | 0.88 | 0.88 | 0.88 | **0.88** |
| pip_pipel | 2259 | 646 | 0.70 | 0.30 | 0.77 | 0.76 | 0.95 | 0.35 | **0.77** |
| sol_ecoli | 700 | 200 | 0.50 | 0.48 | 0.88 | 0.86 | 0.87 | 0.87 | **0.87** |
| toxinpred_trembl | 10041 | 2870 | 0.87 | 0.13 | 0.99 | 0.97 | 1.00 | 0.95 | **0.99** |

focuses on regions enriched with aromatic and cationic residues (F, W, R, and P) within the antimicrobial peptide. Different PRM-enriched antimicrobial peptides contain multiple P and arginine (R) motifs, as described in Huan et al. (2020). R provides both a peptide charge and hydrogen bonding interactions, both of which are important for binding to bacterial membranes Huan et al. (2020). Tryptophan (W) activates arginine-rich regions through ion-pair-pi interactions Walrant et al. (2020). Studies have shown that the molecular size of aromatic residues is crucial in disrupting membrane integrity and folding Strøm et al. (2002); Datta et al. (2016). Figure 2c shows two distinct bands that the model deems structurally important. The sequence is from the V3 loop structure of the HIV-1 R5 subtype, which uses the CCR5 receptor for transmission. Electrostatic properties within the V3 loop are critical for receptor binding and dictate which receptor the virus uses to enter the cell (X4 and R5 within the dataset). Regions 9–12 are especially important because positive charge is associated with efficient binding and the structure of the peptide López de Victoria et al. (2012); Kato et al. (1999). For the marked amino acid (AA) positions 19-23, researchers demonstrated the importance of sulfated tyrosine in the interaction between HIV-1 V3 loops and CCR5 receptors chin Huang et al. (2007). This section likely contributes to the hydrophobic core of the V3 loop, making it important for the structure, as well as for the electrostatic potential differences in the loop that differentiate X4 and R5 tropism, as discussed in Jiang et al. (2010). Figure 2d shows a longer peptide sequence and larger structural areas to which the model pays attention. This is likely because AA changes in larger peptides disrupt the peptide structure less than in smaller peptides Guo et al. (2004); May et al. (2025). However, the model can still identify regions important to the peptide's structure and function. For instance, the Swiss Institute for Experimental Cancer Research used the basic cluster KR-KR-KR to identify neuropeptide AA patterns Nathoo et al. (2001). Further identification marks are the presence of F, P, and L toward the C-terminal of the peptide Agrawal et al. (2019a).

## 4 CONCLUSION

PepTriX improves the performance and interpretability of protein language models (PLMs) by combining 1D ESM-2 embeddings and 3D ESMFold-derived structures within a lightweight graph attention network (GAT). The network is aligned via contrastive learning and explained through cross-modal co-attention via a hybrid loss function. This design avoids costly PLM fine-tuning and reveals biophysically plausible interactions that experts can verify. PepTriX delivers remarkable performance across diverse peptide datasets. It produces task-specific peptide vectors and attention maps that link predictions to structural and physicochemical motifs. The versatile framework can be adapted to various peptide classification tasks, enabling domain experts to use classical techniques for structure generation. However, limitations include a focus on two-class labels and the need for domain expertise to interpret attention. Planned work targets include multi-label and regression settings, as well as calibrated uncertainty. PepTriX is an efficient and interpretable toolkit for peptide classification and structure-guided discovery. For extended discussion, please refer: Appendix A.3

## 5 REPRODUCIBILITY STATEMENT

To support reproducibility, we provide a ZIP file containing an example for running our framework on one of the datasets used in this study. Detailed instructions are available in the *README.md* file.

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

# A  APPENDIX

## A.1  ADDITIONAL RESULTS

Table 2: Here is a comparison between our method and other frameworks or encodings with respect to the best performing $\lambda$. The dataset names in bold are those on which our framework performed better. While other methods focus on specific types of datasets — apart from iCAN, which is an atomic-level, general encoding — our framework is generalizable to all types of datasets.

| Methods | Data set(s) | F1 Score |
|---|---|---|
| cksaap | amp_modlamp Müller et al. (2017), cpp_kelmcpp Pandey et al. (2018), cpp_mlcpp Manavalan et al. (2018b), nep_neuropipred Agrawal et al. (2019b) | 0.93, 0.85, 0.85, 

 0.88 |
| dde | aip_antiinflam Gupta et al. (2017), pip_pipel Manavalan et al. (2018a) | 0.67, 0.56 |
| dist_f | amp_iamp2l Xiao et al. (2013), cpp_cellppd Gautam et al. (2013), cpp_cellppdmod Kumar et al. (2018), cpp_cppredfl Qiang et al. (2018), **cpp_mlcppue** Manavalan et al. (2018b), **cpp_sanders** Sanders et al. (2011b) | 0.82, 0.9, 0.93, 


 0.91, 0.7, 0.89 |
| iCAN | cpp_mixed Dobchev et al. (2010), sol_ecoli Prabakaran et al. (2021), amy_hex Prabakaran et al. (2021), toxinpred_trembl Gupta et al. (2013) | 0.89, 0.72, 0.72, 


 0.77 |
| qsar | **hiv_abc** Löchel et al. (2019), **hiv_apv** Löchel et al. (2019), **hiv_azt** Löchel et al. (2019), **hiv_v3** Dybowski et al. (2010) | 0.97, 0.99, 0.98, 0.99 |
| **Our framework** | **aip_antiinflam** Gupta et al. (2017), **amp_iamp2l** Xiao et al. (2013), **amp_modlamp** Müller et al. (2017), **amy_hex** Prabakaran et al. (2021), hiv_v3 Dybowski et al. (2010), **nep_neuropipred** Agrawal et al. (2019b), **pip_pipel** Manavalan et al. (2018a), **sol_ecoli** Prabakaran et al. (2021), **toxinpred_trembl** Gupta et al. (2013), hiv_abc Löchel et al. (2019), hiv_apv Löchel et al. (2019), hiv_azt Löchel et al. (2019), **cpp_cellppd** Gautam et al. (2013), **cpp_cellppdmod** Kumar et al. (2018), **cpp_cppredfl** Qiang et al. (2018), **cpp_kelmcpp** Pandey et al. (2018), **cpp_mixed** Dobchev et al. (2010), **cpp_mlcpp** Manavalan et al. (2018b), cpp_mlcppueManavalan et al. (2018b), cpp_sanders Sanders et al. (2011b) | 0.70, 0.93, 0.96, 0.83, 



 0.97, 0.90, 0.77, 0.87, 


 0.99, 0.90, 0.89, 0.86, 


 0.91, 0.93, 0.95, 0.86, 


 0.89, 0.85, 0.61, 0.72 |

Different encodings used on those datasets were compared in the paper by Weckbecker et al. (2024), with different ones exceeding on different datasets. For our benchmarking (Table: 2), we selected a range of datasets. The shortest input peptide comprises three amino acids (AA) and originates from the cpp_cellppd dataset, whereas the longest peptide comprises 249 AA and originates from the hiv_abc dataset. With our approach, we either performed comparably or outperformed existing encodings. These results demonstrate the wide applicability of our approach to various biological domains and the ability of our hybrid model of sequence and structural representation to capture the broad biological context and real-world understanding of peptides. Second, we achieved insight into the large-scale protein model ESMFold by leveraging contrastive learning to make the parts of the sequence interpretable that are important to the 3D structure of the investigated peptides. We experimented with different values of the penalizing parameter, lambda, of the hybrid loss function, which penalizes the contrastive loss (Table: 3). The results, F1 scores, for each dataset did not vary much for different values of lambda ($\lambda$). The graph convolutional network in the provided

Table 3: The performance metrics shown here are based on various datasets and include the F1-scores with respect to the different values of the penalizing factor $\lambda$ in the hybrid loss function.

| Dataset | F1 Score | Lambda($\lambda$) |
|---|---|---|
| aip_antiinflam | 0.68 | 0.01 |
| aip_antiinflam | 0.69 | 0.1 |
| aip_antiinflam | 0.70 | 0.5 |
| amp_iamp2l | 0.92 | 0.01 |
| amp_iamp2l | 0.92 | 0.1 |
| amp_iamp2l | 0.93 | 0.5 |
| amp_modlamp | 0.96 | 0.01 |
| amp_modlamp | 0.95 | 0.1 |
| amp_modlamp | 0.94 | 0.5 |
| amy_hex | 0.83 | 0.01 |
| amy_hex | 0.81 | 0.1 |
| amy_hex | 0.79 | 0.5 |
| hiv_v3 | 0.96 | 0.01 |
| hiv_v3 | 0.97 | 0.1 |
| hiv_v3 | 0.96 | 0.5 |
| nep_neuropipred | 0.89 | 0.01 |
| nep_neuropipred | 0.88 | 0.1 |
| nep_neuropipred | 0.90 | 0.5 |
| pip_pipel | 0.77 | 0.01 |
| pip_pipel | 0.77 | 0.1 |
| pip_pipel | 0.77 | 0.5 |
| sol_ecoli | 0.87 | 0.01 |
| sol_ecoli | 0.85 | 0.1 |
| sol_ecoli | 0.86 | 0.5 |
| toxinpred_tremble | 0.97 | 0.01 |
| toxinpred_tremble | 0.94 | 0.1 |
| toxinpred_tremble | 0.99 | 0.5 |

code is a two-layer Graph Attention Network (GAT) that processes the 3D structural information of peptides. This architecture consists of two sequential GATConv layers, each of which utilizes a single attention head. The network maintains a uniform feature dimension of 1280 throughout its layers, a value determined by the configuration of the ESM2 encoder. A Gaussian Error Linear Unit (GELU) activation function is applied between the two layers to introduce nonlinearity, enabling the model to learn complex spatial relationships by gathering information from neighbors up to two hops away. We did not extensively tune hyperparameters of the GAT architecture, such as the number of layers, hidden dimensions, or attention heads, since our goal was to keep the model compact to enhance interpretability and to achieve strong performance without incurring substantial computational cost. In our framework, we used the AdamW optimizer Loshchilov & Hutter (2017) with an exponential learning rate scheduler. AdamW provides robust, decoupled weight decay for regularization. This prevents overfitting and improves generalization in learned peptide embeddings. Meanwhile, the exponential scheduler steadily reduces the learning rate with each epoch. This enabled rapid early training and fine-tuned updates as clustering improved. Training was performed for up to ten epochs. An early stopping callback halts training if the validation loss does not decrease for three consecutive epochs. This ensures computational efficiency and prevents overfitting. We didn't experiment much with different learning rate schedulers or optimizers because they were outside the scope of our research.

## A.2 ADDITIONAL INTERPRETABILITY ANALYSIS

Figure 3 illustrates the attention patterns for four peptides from the pip_pipel dataset, contrasting positive (proinflammatory) and negative (non-proinflammatory) classes. As shown in Figure 3a and Figure 3b, attention maps for positively identified peptides exhibit distinct patterns, whereas the negative cases in Figure 3c and Figure 3d lack such structure. This outcome is expected because the model learns structural representations corresponding to biological function, in this case proinflammatory induction, while no specific structural motif defines non-proinflammatory peptides, resulting in diffuse or uniform attention for negative samples. In Figure 3a, the model highlights residues con-

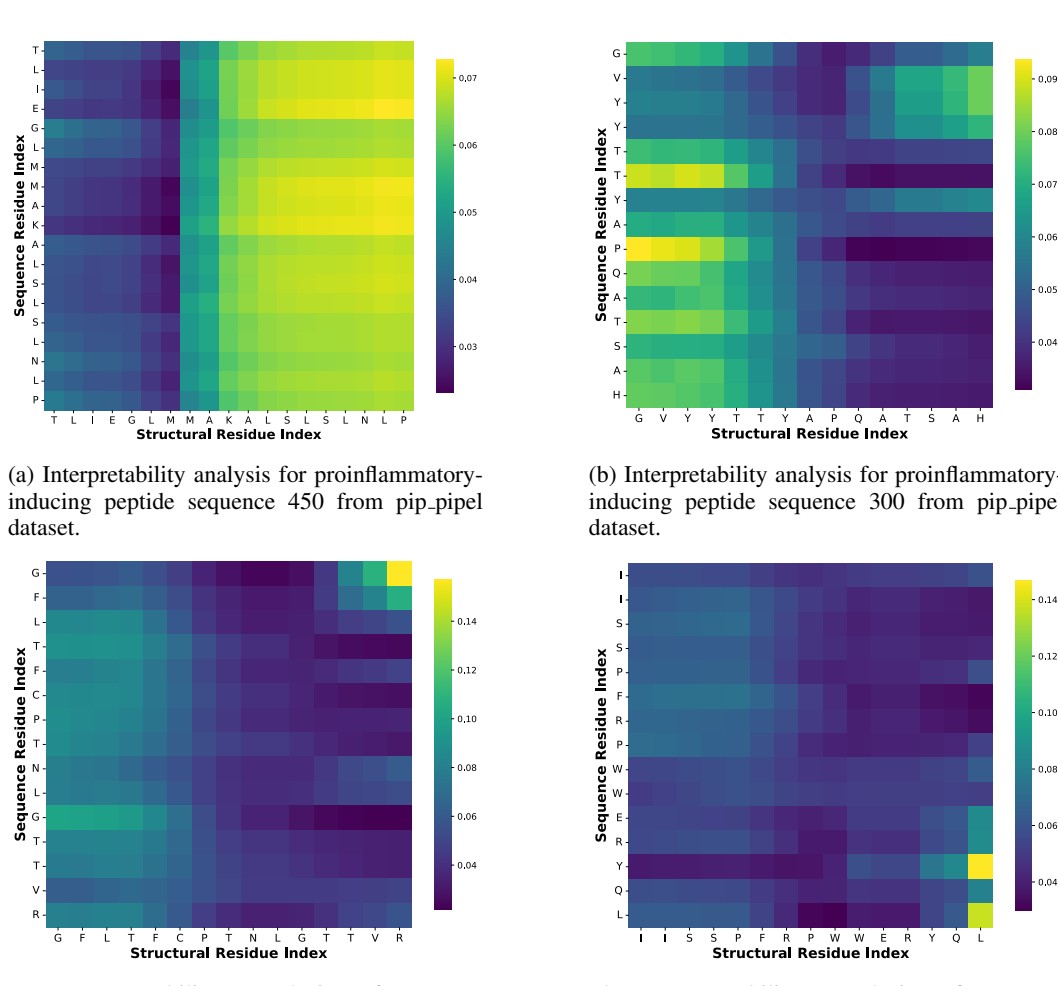

(a) Interpretability analysis for proinflammatory-inducing peptide sequence 450 from pip_pipel dataset.

(b) Interpretability analysis for proinflammatory-inducing peptide sequence 300 from pip_pipel dataset.

(c) Interpretability analysis for non-proinflammatory-inducing peptide sequence 320 from pip_pipel dataset.

(d) Interpretability analysis for non-proinflammatory-inducing peptide sequence 1266 from pip_pipel dataset.

Figure 3: Interpretability analysis for four sequences from the pip_pipel dataset. Two sequences were successfully identified by the model as proinflammatory-inducing and two as non-proinflammatory-inducing. The heatmaps show the attention matrix learned by the classifier, with rows corresponding to sequence residues and columns to structural residue indices. Brighter colors show higher attention and darker colors lower attention.

sistent with known residues of proinflammatory peptides, such as leucine (L), which may facilitate binding to Major Histocompatibility Complex (MHC) molecules, an important component of the immune response Gupta et al. (2016). In contrast, the sequence in Figure 3b shows attention focused on residues not widely described in the literature, including glycine, valine, tyrosine, tyrosine (GVYY), and proline (P). These residues may contribute to critical three-dimensional conformations and represent potentially rare motifs associated with proinflammatory activity.

## A.3  EXTENDED DISCUSSION

The emergence of large-scale protein language models (PLMs), such as ESM-2 and its structure-predicting counterpart ESMFold, has been revolutionary in the disciplines of bioinformatics and drug discovery. Trained on extensive databases of protein sequences, these models can generate robust embeddings that encapsulate significant evolutionary and biophysical information. However, when applied to specialized peptide classification tasks, a significant paradox emerges. Although

they have immense predictive potential, their practical utility is limited by critical issues of cost and clarity. This hinders the translation of computational predictions into domain-centric insights applicable to researchers.

The most salient obstacle pertains to the substantial computational expense associated with these models. Refining a model with a large number of parameters, such as ESM-2, for a specific task, such as predicting peptide toxicity, requires substantial computational resources. Accessing high-end GPU clusters for extended periods is often beyond the financial and logistical capacity of academic research groups and smaller biotech companies, posing a significant challenge. This "computational fortress" has the unintended consequence of impeding innovation by constraining researchers' capacity to swiftly prototype, evaluate novel hypotheses, or utilize these state-of-the-art tools in niche datasets. Despite the availability of resources, PLMs encounter a significant interpretability challenge, resulting in their operation as a "black box." While it is possible to envision the internal attention maps of models such as ESMFold, these maps are intricate webs of scores from numerous layers and hundreds of attention heads. These models are optimized for a general pre-training objective, such as predicting a masked amino acid. They are not inherently structured to answer a biologist's specific question, such as "Which residues form the toxic motif?" The process of deciphering these patterns to extract clear, biologically relevant insights constitutes a complex post-hoc analysis task, often yielding ambiguous results. This lack of transparency is a significant hindrance to drug discovery, where understanding the mechanism of action and the rationale behind a prediction is as important as the prediction itself. The present study proposes PepTriX, a framework designed to facilitate interpretable, efficient, and domain-centric peptide science.

The PepTriX framework was designed to systematically reduce these limitations, providing a computationally feasible solution rich in interpretable, domain-specific insights. Rather than substituting the capabilities of PLMs, the system intelligently leverages and enhances them. PepTriX builds a direct path from raw data to actionable biological understanding by strategically combining sequence embeddings, 3D structural data, and a purpose-built neural architecture. PepTriX's fundamental approach eliminates the need for costly fine-tuning, addressing prevailing cost concerns. It utilizes the advanced ESM-2 model as a feature extractor to generate pre-computed 1D sequence embeddings in a one-time offline step. The actual learning is then performed by a comparatively much lighter and more efficient Graph Attention Network (GAT).This approach democratizes access to state-of-the-art representational power, allowing for rapid model training on standard hardware. The core impact of PepTriX lies in its ability to address the interpretability challenge. The GAT is well-suited for modeling 3D peptide structures by treating amino acids as nodes and their spatial proximity as edges. Unlike the generic attention mechanisms inherent to a PLM, the GAT's attention mechanism is specifically trained for the classification task at hand. This suggests that the model learns to allocate high attention scores to the residues and structural motifs that are causally significant for the predicted property (e.g., HIV inhibition). This approach yielded a clear, quantitative map of the peptide's functional hotspots. Furthermore, PepTriX improves upon this structural analysis with two key innovations. First, cross-modal co-attention functions as a unifying bridge, compelling the model to learn the intricate interplay between the one-dimensional (1D) sequence context (from ESM-2) and the three-dimensional (3D) structural conformation (from the GAT). For example, the system can determine the importance of a glycine in the sequence because of its role in enabling a specific conformational change in the three-dimensional structure, which is necessary for effective binding. To ensure the model's predictions are robust and specific, contrastive training identifies differences in a highly specialized way. The model is explicitly taught to group similar peptides (e.g., two cell-penetrating peptides) together and separate dissimilar ones in the embedding space. This process refines the general-purpose embeddings into a highly specialized vector space, tailored precisely to the classification task at hand.

PepTriX demonstrates noteworthy and frequently remarkable performance across a range of peptide classification tasks, including toxicity prediction, HIV inhibition, antimicrobial peptide identification, and cell-penetrating peptide identification. However, its most significant contribution is its ability to bridge the gap between correct predictions and useful discoveries. The framework provides more than just a classification label; it also offers interpretable validation that highlights the specific structural and biophysical drivers behind its decisions. For domain experts, this transformation of the model from a black-box predictor to an insightful research tool enables hypothesis generation, informed peptide design, and accelerated progress from computational hits to validated therapeutic candidates.

