# OpenReview forum: "PEPTRIX: A FRAMEWORK FOR EXPLAINABLE PEPTIDE ANALYSIS THROUGH PROTEIN LANGUAGE MODELS"
_ICLR.cc/2026/Conference — ICLR 2026 Conference Withdrawn Submission_

### Official Review · Reviewer_PayV · 2025-10-31

**Soundness:** 2
**Presentation:** 1
**Contribution:** 1
**Rating:** 2
**Confidence:** 4

**Summary:**

The work presents a new model for peptide property prediction called PepTriX. PepTriX uses ESM2 and ESM2-fold to encode 1D and 3D information of peptides, and uses graph convolutions and contrastive learning for property prediction. The model is evaluated on multiple peptide property prediction tasks, including antimicrobial activity, toxicity, and solubility, and a domain expert interprets several predictions.

I am recommending this paper to be rejected in its current form for the following reasons:

1. The model needs more validation to support state-of-the-art claims.
2. The interpretability section needs to be strengthened.
3. The writing and organization need a major rework.

**Strengths:**

1. A new model is developed that combines 1D and 3D information of peptides for property prediction.
2. Expert analysis is provided for some predictions.

**Weaknesses:**

1. The model is tested on multiple datasets, but it is difficult to compare with prior work. Are the same train/test splits used for the baselines? Are hyperparameters tuned on a validation set? What is the dissimilarity between train/test splits? More details are needed on the experimental setup to support state-of-the-art claims, or to estimate the validity of the findings.
2. While expert analysis is a nice first step towards interpretability, a global analysis is needed to claim that the model is capturing patterns relevant to the task. The current analysis remains anecdotal.
3. The difference between the tasks is understudied. It is expected that different tasks require capturing different patterns. An analysis of when each representation (1D, 3D, combined) is more useful is essential for a paper that combines multiple representations.
4. The writing and organization should be redone. At the moment, the work contains around seven pages of background and methods, and around one and a half pages of results. Yet, key details on experimental setup and analysis are missing. I recommend condensing the first seven pages and concisely incorporating more experiments and discussions to enhance the paper's clarity and readability.

Small comments (not reason for rejection, but should be addressed in a revision):
1. The citations format should be consistent (brackets around citations). ICLR has a specific format that should be followed.
2. L424: "Figure fig:all" is a broken LaTeX reference.

**Questions:**

1. Are the baselines and PepTriX trained and evaluated on the same train/test splits? How are hyperparameters selected for all models?
2. What is the dissimilarity between train/test splits for each dataset?
3. What is the impact of each component of the model (ESM2, ESM2-fold, GCN, contrastive learning) on performance? An ablation study is needed.
4. Can the authors provide a global analysis of interpretability, beyond the expert analysis of a few examples?
5. Can the authors provide an analysis of when each representation (1D, 3D, combined) is more useful for prediction, across tasks?

---

### Official Review · Reviewer_oAGq · 2025-10-31

**Soundness:** 2
**Presentation:** 1
**Contribution:** 1
**Rating:** 2
**Confidence:** 4

**Summary:**

This paper addresses the problem of interpretable peptide classification and proposes a multimodal method that integrates and aligns the embeddings generated by the ESM-2 PLM and the accompanying ESMFold model, respectively, for peptide sequence and structure. A contrastive loss is used for the multi-modal alignment, and a cross-attention mechanism is employed to interpret the interplay between the two modalities when driving the downstream predictions.

**Strengths:**

- Interpretation and explanation of deep neural networks, especially foundation models, in peptide analysis is an important and timely problem of interest to the ICLR audience.
- The biological explanations of the derived attention maps are interesting (though unclear whether they are generalizable).

**Weaknesses:**

- It is somewhat unclear what the specific contribution of the paper is. As written, the paper primarily serves as a lengthy review of background and related work, making it challenging to understand where the proposed method begins. Most of the content before pages 6-7 could be moved into a background or related work section, allowing the proposed method to be emphasized in a dedicated section. Concepts such as attention, graph attention network, and contrastive learning are familiar enough to an average ICLR reader that they do not require as much space in the paper.
- It is unclear if and how the attention is applied to the two modalities, and which modality is actually driving the downstream prediction. Is the cross-attention applied at the token-level and before the contrastive learning calculation? Which modality (sequence or structure, or potentially both) is used for the downstream classification?
- Following on the above comment, how are $z$'s defined in Eq. (5)? Are these embeddings at the amino acid (AA) level or at the peptide level? In the latter case, is there a pooling mechanism involved?
- There are many long paragraphs across the paper (such as the first two paragraphs of the introduction section and the one after Eq. (5)-(6)) that need to be broken down to be more readable.

**Questions:**

- The authors mention the limitation on the need for domain expertise for evaluating explainability. I agree with that, and since this is the primary premise of the paper, it warrants a more extensive discussion on how the explanations provided by the proposed method could be evaluated more broadly, potentially through the systematic involvement of biologists and domain experts.
- The performance metrics reported in Table 1 are hard to evaluate at an absolute level, and there needs to be a comparison with other baselines. I suggest that the results from Appendix A.1 be moved to Table 1 to better put them into perspective with respect to prior work.

---

### Official Review · Reviewer_cSyN · 2025-11-01

**Soundness:** 2
**Presentation:** 2
**Contribution:** 2
**Rating:** 0
**Confidence:** 5

**Summary:**

This authors proposed PepTriX, a multimodal framework that combines 1D sequence embeddings from ESM-2 with 3D structure features (ESMFold) processed by a GAT, trained with a hybrid loss that blends task classification and contrastive alignment across modalities. A cross-modal co-attention block provides residue-level interpretability linking sequence tokens to structural neighborhoods. The authors benchmark on nine binary peptide tasks (AMPs, toxicity, HIV V3 tropism, etc.), reporting strong F1 on several datasets (e.g., 0.96 on amp_modlamp; 0.99 on toxinpred_trembl) and qualitative heatmaps that highlight biophysically plausible motifs (e.g., aromatic/cationic regions for AMPs, basic clusters in HIV V3). The paper argues that PepTriX addresses two persistent issues with PLMs by avoiding heavy PLM fine-tuning and by exposing cross-modal attention maps for expert validation.

**Strengths:**

The paper have clear probelm framing andprincipled architecture coupling 1D PLM embeddings with 3DGAT, aligned by constractive learning. Interpretable co-attention that links residue tokens to structural neighborhoods; examples match biophysical expectations (AMP aromatic/cationic residues; HIV V3 electrostatics).

**Weaknesses:**

Table 1 lacks standardized, like-for-like comparisons versus strong PLM or structure-aware baselines trained under identical data splits and budgets; limited ablation of each module’s contribution (GAT vs. 1D only; contrastive vs. none; co-attention vs. none). Qualitative heatmaps are compelling but need systematic human-expert studies (blinded scoring, inter-rater reliability) or quantitative localization metrics (e.g., enrichment near known motifs/contacts).Heavy reliance on ESMFold structures; impact of structure noise/confidence not analyzed (e.g., filtering by pLDDT, sensitivity to misfolds). Focus on binary classification; multi-label/regression and uncertainty calibration deferred to future work.

**Questions:**

1. Many prior frameworks are task-specific; in your selection of nine datasets, how did you ensure diversity (length, composition, structure content) that truly stresses generalization rather than overlapping inductive biases?
2. Interpretability is a key selling point; what downstream decisions (e.g., peptide redesign rules) did your explanations concretely enable in expert workflows? Any examples beyond visual inspection?
3. What is performance with 1D-only (ESM-2 + classifier), 3D-only (GAT on ESMFold), and no-contrastive settings? Please report F1 deltas and explanation fidelity changes.
4. How sensitive is PepTriX to ESMFold confidence? Did you stratify results by pLDDT or introduce uncertainty-aware weighting into the GAT/co-attention?
5. How stable are results under class imbalance, sequence length extremes, and ESMFold errors? Any failure modes where explanations look plausible but predictions are wrong?

---

### Official Review · Reviewer_9rR3 · 2025-11-01

**Soundness:** 2
**Presentation:** 3
**Contribution:** 2
**Rating:** 2
**Confidence:** 4

**Summary:**

The paper introduces PepTriX, a framework for peptide classification (e.g., predicting toxicity, antimicrobial activity) that aims to deliver both high performance and interpretability, while avoiding the high computational costs of fine-tuning large Protein Language Models (PLMs). The method employs a dual-encoder architecture. First, it uses pre-trained ESM-2 embeddings to represent the 1D peptide sequence. Second, it generates a 3D structure using ESMFold, which is then processed by a Graph Attention Network (GAT) to create a structural representation. These two (1D sequence and 3D structure) representations are aligned using a contrastive learning objective. The entire model is trained with a hybrid loss function that combines this contrastive loss with a standard classification loss. The framework's claim to interpretability is based on a cross-modal co-attention module, which generates heatmaps intended for validation by domain experts.

**Strengths:**

1. **Important Problem:** The paper tackles a critical and relevant problem: creating generalizable, computationally efficient, and interpretable models for peptide function prediction.
2. **Comprehensive Related Work:** The authors provide a thorough review of existing methods for peptide classification and interpretability, clearly positioning their work within the current landscape.
3. **Logical Framework:** The core idea of aligning a 1D sequence "blueprint vector" with a 3D structural "summary vector" via contrastive learning is intuitive and biochemically sound.
4. **Generalizability:** The framework is benchmarked on nine different datasets covering a wide range of biological functions (antimicrobial, HIV, toxicity, etc.), which demonstrates a good degree of generalizability.

**Weaknesses:**

1. **Limited Novelty:** The main weakness of this paper is its lack of originality. The framework is an assembly of standard parts: pre-trained ESM-2 embeddings, ESMFold for structure generation, a GAT for processing graphs, contrastive loss for modal alignment, and co-attention for heatmaps. The paper fails to prove that this specific combination provides a unique advantage that other, simpler combinations would lack.
2. **Weak and Self-Contradictory Interpretability Claim:** The paper's primary contribution is purported to be interpretability. However, it criticizes other models' attention/saliency maps as potentially misleading but then offers its own co-attention heatmaps as a solution without any rigorous validation of their "faithfulness." The analysis in Section 3 is purely qualitative and prone to the exact post-hoc rationalization that plagues attention-based interpretability.
3. **Overstated Performance and Cost Claims:** The claim to "match or exceed" SOTA is not supported by the data in Table 2, which shows mixed results. Furthermore, the claim of being "lightweight" by avoiding fine-tuning is a strawman. The proposed method *replaces* fine-tuning with the non-trivial computational cost of running ESMFold inference for every peptide and then training a GAT on thousands of individual graphs. A much simpler and truly lightweight baseline (e.g., an MLP trained *only* on the ESM-2 embeddings) is not compared against, making it impossible to judge the utility of the entire 3D/GAT branch.
4. **Unexplored Hybrid Loss:** The hybrid loss simultaneously optimizes for classification and seq-struct alignment. It is not demonstrated that this auxiliary contrastive task actually *helps* the primary classification task. The results in Table 3, showing that model performance is highly *insensitive* to the $\lambda$ weight, are puzzling. This could imply the contrastive component has no meaningful impact, rendering a key part of the proposed method unnecessary. This is a critical point that is left unaddressed.

**Questions:**

1. **On Interpretability:** The paper criticizes other attention mechanisms as potentially misleading. How does the proposed co-attention mechanism avoid these same pitfalls? Can the authors provide *quantitative* validation of their heatmaps' faithfulness (e.g., via-label-erasure experiments) rather than relying on qualitative, post-hoc stories?
2. **On Novelty & Utility:** Given that the components (ESM embeddings, GATs, contrastive loss) are all standard, what is the core novel insight? Specifically, can the authors provide an ablation study that compares the full PepTriX model to a much simpler baseline, such as a 2-layer MLP trained *only* on the ESM-2 embeddings? This would be necessary to justify the added complexity and computational cost of the entire ESMFold/GAT structural branch.
3. **On the Hybrid Loss:** The results in Table 3 show that performance is remarkably stable across different values of $\lambda$. This is counter-intuitive and suggests the contrastive loss $\mathcal{L}_{contrastive}$ might not be contributing meaningfully to the training. Could the authors elaborate on this? What happens if $\lambda=0$? Is the performance identical? If so, this would invalidate the premise of the hybrid-loss approach.
4. **On Performance Claims:** In light of the mixed results in Table 2 (e.g., `hiv_v3` baseline is 0.99 F1 vs. PepTriX's 0.97), would the authors consider toning down the claims of "remarkable performance" 44 to a more accurate "comparable performance"?

---

### Official Review · Reviewer_u2D6 · 2025-11-01

**Soundness:** 3
**Presentation:** 2
**Contribution:** 2
**Rating:** 2
**Confidence:** 4

**Summary:**

The paper proposes PepTriX, a framework integrating ESM-2 sequence embeddings (1D) with ESMFold-derived 3D features via a Graph Attention Network (GAT). The model uses a contrastive loss to align sequence and structure embeddings, and introduces a cross-modal co-attention module for interpretability. Empirical results includes multiple peptide classification tasks.

**Strengths:**

The main strength of the paper is that the approach is strongly motivated by biologically relevant questions.

1. This paper is strongly motivated by the underlying biological problems which addresses the interpretability crisis in theses models.
2. The method leverages multi modal input integrating 1D and 3D features generated by in silico oracles.
3. Strong empirical evidence with large number of datasets tested.

**Weaknesses:**

The main weakness of this model is the novelty and the lack of systemic analysis of the results beyond anecdotal observation.

1. This method mainly compose of existing methods, no material novel insights regarding the rationale for the design choices. This paper will hugely benefit from a more principled study of why and how the design choices are made and logical reasoning connecting them back to motivation.
2. The results are mostly anecdotal, no benchmarking on interpretability metrics such as saliency agreement. At least an attribution over-lap with known functional structure motif should be shown to enhance the results section.
3. The lack of ablation study makes it hard to examine how each of the component plays into the final results.

**Questions:**

1. How sensitive are the attributions with sequence mutation and folding methods?
2. How does this generalize cross datasets where the paper claimed domain transferability?
3. How's the 3D graph constructed from the predicted structures?

---

### Note · Authors · 2025-11-13

**Comment:**

We appreciate the detailed feedback provided by the reviewers.
After internal discussion, we have decided to withdraw the manuscript at this time.

Thank you for your consideration.

**Withdrawal Confirmation:**

I have read and agree with the venue's withdrawal policy on behalf of myself and my co-authors.